

# Titanium dioxide dental implants surfaces related oxidative stress in bone remodeling: a systematic review

Elaf Akram Abdulhameed[1,2], Natheer H. Al-Rawi[3], Marzuki Omar[1], Nadia Khalifa[2] and A.B. Rani Samsudin[3]

[1] School of Dental Sciences, Universiti Sains Malaysia, Kelantan, Malaysia
[2] Preventive and Restorative Dentistry, College of Dental Medicine, University of Sharjah, Sharjah, United Arab Emirates
[3] Oral and Craniofacial Health Sciences, University of Sharjah, Sharjah, United Arab Emirates

## ABSTRACT

**Background**. Titanium dioxide dental implants have a controversial effect on reactive oxygen species (ROS) production. ROS is necessary for cellular signal transmission and proper metabolism, but also has the ability to cause cell death as well as DNA, RNA, and proteins damage by excessive oxidative stress. This study aimed to systematically review the effect of titanium dioxide dental implant-induced oxidative stress and its role on the osteogenesis-angiogenesis coupling in bone remodeling.

**Methods**. This systematic review was performed conforming to preferred reporting items for systematic review and meta-analysis (PRISMA) model. Four different databases (PubMed, Science Direct, Scopus and Medline databases) as well as manual searching were adopted. Relevant studies from January 2000 till September 2021 were retrieved. Critical Appraisal Skills Programme (CASP) was used to assess the quality of the selected studies.

**Results**. Out of 755 articles, only 14 which met the eligibility criteria were included. Six studies found that titanium dioxide nanotube (TNT) reduced oxidative stress and promoted osteoblastic activity through its effect on Wnt, mitogen-activated protein kinase (MAPK) and forkhead box protein O1 (FoxO1) signaling pathways. On the other hand, three studies confirmed that titanium dioxide nanoparticles (TiO$_2$NPs) induce oxidative stress, reduce ostegenesis and impair antioxidant defense system as a significant negative correlation was found between decreased SIR3 protein level and increased superoxide (O$_2\bullet$-). Moreover, five studies proved that titanium implant alloy enhances the generation of ROS and induces cytotoxicity of osteoblast cells via its effect on NOX pathway.

**Conclusion**. TiO$_2$NPs stimulate a wide array of oxidative stress related pathways. Scientific evidence are in favor to support the use of TiO$_2$ nanotube-coated titanium implants to reduce oxidative stress and promote osteogenesis in bone remodeling. To validate the cellular and molecular cross talk in bone remodeling of the present review, well-controlled clinical trials with a large sample size are required.

Corresponding author
Elaf Akram Abdulhameed,
Ealzubaidi@sharjah.ac.ae,
elaf.alzubaidi@gmail.com

## INTRODUCTION

Injury, infections and malignancies of maxillofacial region cause defects in hard and soft tissues. Small defects heal on their own in healthy people, while big defects require scaffolded dental implant to allow for sufficient hard and soft tissue regeneration (*Zeng et al., 2018*). Diagnostic imaging technology has witnessed a giant revolution in the last three decades. With the introduction of 3D imaging, the hard tissue regeneration can be visualized in all planes rather than using a two-dimensional evaluation (*Hans, Martin Palomo & Valiathan, 2015*). CBCT evaluation is one of the non-invasive, well suited method to analyze and evaluate bone texture and regeneration on a valuable modality that precisely evaluates skeletal components in the craniofacial region with a 1:1 image (no magnification) (*Alhammadi et al., 2021*). However, it is of limited value in the assessment of soft tissue facial characteristics (*Ludlow et al., 2007*; *Kitai et al., 2017*).

Many studies (*Gomes et al., 2013*; *Mehta, Sagarkar & Mathew, 2017*) have evaluated the potential correlation between craniofacial measurements obtained from the gold standard cephalometric adiographs and analogous measurements from standardized facial profile photographs. They found the standardized photographic method to be repeatable and reproducible. Further, they considered it to be a feasible and practical non-invasive alternative diagnostic method so long as the standardized protocol is followed. Another study concluded that the soft tissue analysis on photographs is a reliable method to evaluate the soft tissue profile compared to the analyses performed on cephalograms (*Nucera et al., 2017*).

Dental implants are foreign bodies that when implanted in bone tissue will lead to generation of Reactive Oxygen Species (ROS). ROS in physiological amount is needed for cellular signal transduction and physiological metabolism. However when the generation of ROS exceed the antioxidant capacity, oxidative stress develop and normal tissue haemostasis imbalance occur leading to poor tissue regeneration and wound healing (*Lee et al., 2021*).

The reactive oxygen species (ROS) is an unstable oxygen-containing molecule that interacts with other molecules in the cell. Its reaction has the ability to cause cell death as well as DNA, RNA, and proteins damage (*Srinivas et al., 2019*). ROS is necessary for cellular signal transmission and proper metabolism in physiological quantities. Oxidative stress occurs when ROS production exceeds antioxidant capacity, disrupting normal tissue homeostasis and resulting in poor tissue regeneration and wound healing (*Huang et al., 2021*).

Bone is a metabolically active structure because it undergoes continual remodeling throughout life. This remodeling occurs as a result of bone resorption and deposition. Biomolecules called Bone Turnover Markers (BTMs) are released into the blood during bone resorption and deposition (*Carey, Licata & Delaney, 2006*). The presence of the dental implants in oral tissues greatly alters their function, this is visible through the analysis of the inflammation markers present, in a study performed by *Guarnieri et al. (2021)* to compare gingival tissue healing at surgically manipulated periodontal sites and at sites receiving implants and healing abutments with machined (MS) vs laser-microtextured (LMS) surface placed with one-stage protocol. He concluded that both MS and LMS

implant sites presented a higher pro-inflammatory state in the early phase after surgery (1–4 weeks). At 12 weeks, only MS implant sites kept a higher pro-inflammatory state, while at LMS implant sites, it becomes similar to surgically and non-surgically manipulated periodontal tissues (*Guarnieri et al., 2021*).

Bone resorption process takes around ten days to take place, whereas bone formation takes around two months to heal or replace the deficiency. As a result, osteoclastic activity is faster than osteoblastic activity, posing a challenge to surgeons and scientists when it comes to bone and skeletal repair and regeneration (*Mizuno & Glowacki, 2000*).

The formation of new blood vessels is critical for bone metabolism, modeling, and remodeling, particularly in osteogenesis and bone repair, as seen in bone fracture healing, for example (*Kanczler & Oreffo 2008*; *Santos & Reis, 2010*). Angiogenesis, which is the sprouting of new arteries from pre-existing ones after activation of endothelial and vessel wall stem cells, and osteogenesis, which is the induction of progenitor cells into the osteoblast lineage, are closely coupled. An autocrine and paracrine network of factors generated by osteoblasts, endothelial cells, and their progenitors regulates the interaction between osteogenesis and angiogenesis (*Riddle et al., 2009*).

Under physiological settings, regulatory proteins and proper signal transductions closely govern all these components that orchestrate bone repair. When bone cells are exposed to oxidative stress which are released as a result of damage, bacterial toxins, or bone augmentation in bone grafting surgery procedures, ROS may impede or undermine the complicated bone regeneration process (*Wauquier et al., 2009*) (Fig. 1).

Dental implants are commonly used in dentistry, orthopedic surgery and other specialties that work on human skeleton (*Sayed et al., 2021*). Biocompatibility properties of these materials have been studied extensively. However, there is debate over the function of ROS in the remodeling process and the dental implants' long-term stability. The key function of ROS in angiogenesis-osteogenesis coupling may influence the effectiveness of dental implant osseointegration because bone is a highly vascularized tissue (*Hu et al., 2018*). There has recently been a growing body of evidence demonstrating the link between ROS generation during intraosseous dental implant implantation and bone vascularization and remodeling. The primary objective of this systematic review is to estimate the extent of ROS release after dental implant placement in bone and its impact on bone remodeling.

## MATERIALS AND METHODS

### Protocol registration
This systematic review was conducted with the standard regulations of Preferred Reporting Items for Systematic Reviews and Meta-Analysis (PRISMA) statement. It is registered with the appropriate guideline protocol with the International Prospective Register of Systematic Reviews (PROSPERO) platform (ID: CRD42021271610).

### Focused question
Does dental implants-induced oxidative stress have an effect on bone remodeling?

The current systematic review was adopted to follow PICO criteria:

P: Patient who underwent biomaterial placement using either nanoparticles or nanotube.

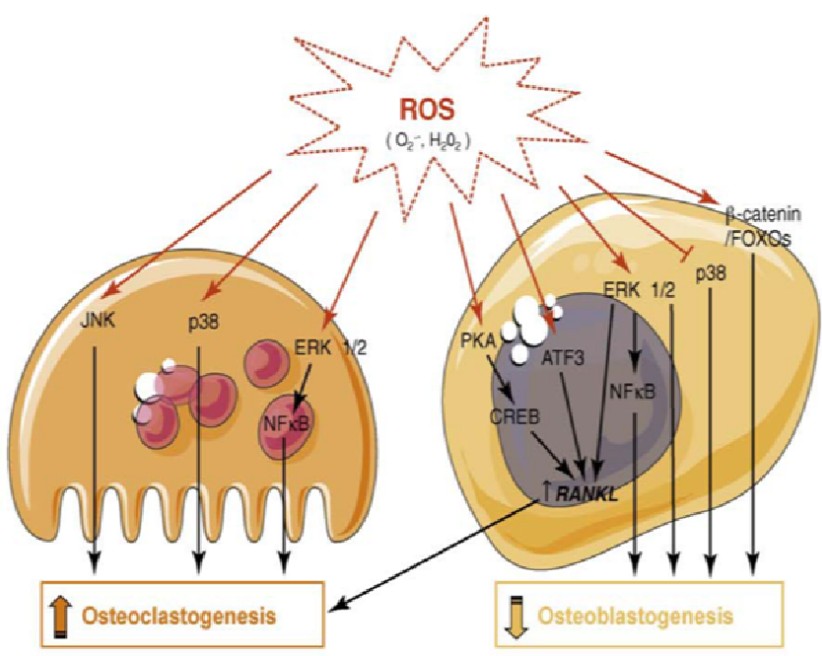

**Figure 1** **ROS modulation of signaling pathways in bone cells.** ROS promote bone loss by inhibiting osteoblast differentiation and enhancing osteoclastogenesis. ROS induced bone resorption occurs directly or indirectly (increased RANKL expression) through the modulation of kinases and transcription factor activities in both osteoclasts and osteoblasts. Figure reprinted from Trends in Molecular Medicine, Vol 15, Wauquier F, Leotoing L, Coxam V, Guicheux J, Wittrant Y., Oxidative stress in bone remodelling and disease, 2009 Oct;15(10):468-77. doi: 10.1016/j.molmed.2009.08.004.

I: Intervention; two different biomaterials were used nanoparticles and nanotube.
C: Comparison on the amount of oxidative stress release from the two biomaterials.
O: Oxidative stress release amount from the two biomaterials.

## Search strategy

Four databases were used for this systematic review (PubMed, Science Direct, Scopus and Medline). The search included the following sets of key words like: (Oxidative stress) or (Reactive oxygen species); (Titanium dioxide nanoparticles); (Titanium dioxide nanotube); (Titanium alloy); (Osteogensis); (Angiogenesis) or (VEGF). The mesh term used was (Titanium dioxide nanoparticles AND oxidative stress); (Titanium dioxide nanotubes AND oxidative stress); (Titanium alloy AND oxidative stress); (Bone regeneration AND oxidative stress), (Oxidative stress AND bone regeneration); (Titanium dioxide nanoparticles, oxidative stress AND osteogenesis-angiogenesis coupling); (Titanium dioxide nanotubes, oxidative stress AND osteogenesis-angiogenesis coupling); (Dental implant, oxidative stress AND osteogenesis-angiogenesis coupling).

The search terms employed were key words classified under the general (all fields) category. The search terms were combined with an 'OR' and categories were combined using 'AND' or 'NOT' to create a final search query. The following filters were applied to these terms: Full text, published in the last 21 years (since 2000), English and academic

literatures only. Two independent reviewers (EAA and NHA) conducted the search from April 2021 to September 2021.

## Inclusion and exclusion criteria
### Inclusion criteria
Full text papers published in English literature. All clinical investigations on oxidative stress, dental implant, bone regeneration, and osteogenesis-angiogenesis coupling that were conducted *in vivo*, *in situ*, *in vitro*, or in cell culture met the inclusion criteria for the period from January 2000 till September 2021.

### Exclusion criteria
Studies that did not highlight the influence of oxidative stress on dental implant, bone regeneration, or osteogenesis-angiogenesis coupling were excluded. Studies that did not look at cross talk in the context of bone remodeling in the presence of oxidative stress were also eliminated.

## Eligibility criteria
In order to obtain precise results, clinical studies and clinical trials were included in this study. Papers that have been peer reviewed were selected as well. The aim of those selected studies is to figure out whether dental implant-induced oxidative stress influence the osteogenesis-angiogenesis coupling in bone remodeling. Articles involving expert's opinions were excluded. Abstract in conferences, and letters to editors found in articles were also excluded, as well as non-peer reviewed and non-English papers.

## Data extraction
The papers selected from four databases (PubMed, Science Direct, Scopus and Medline) were reviewed by two authors separately (EAA and NHA). This procedure was completed through the following steps. First, after selecting the papers from database search, duplicated papers were removed manually. Secondly, each person read through the abstract of the paper, selecting them based on the inclusion and exclusion criteria. The final step included reading full text paper. Screening was conducted by two authors independently (EAA and NHA.) and any disagreement was resolved via discussion with a third reviewer (RS) to reach a consensus. Level of inter-reviewer agreement was determined by Cohen Kappa score. An article (*McHugh, 2012*) the eligibility criteria was included for quality. Then, data were extracted from each one including name of author/year, aim of the study, design, sample characteristics, interventions, and assaying OS levels.

## Studies quality assessment
The Critical Appraisal Skills Programme (CASP) (Long, French & Brooks, no date) assessed the quality of the included studies. The CASP tool is a generic tool for appraising the strengths and limitations of any qualitative research paper. To assess the quality of the study, three different domains (Introduction, methodology, results and discussion) was used. A tick scoring system was used for each study. In CASP, there are 30 questions for the three domains, each question score 3.33. The studies were graded as ''Strong,''

"Moderate", and "Weak" based on the total CASP score. Studies that scored less than 33% were considered weak, between 33% and 67% were considered moderate and higher than 67% were considered an article of strong evidence.

## Analysis

A descriptive summary of the findings are tabulated based on the focused question as seen in Table 1. CASP score was used to assess the quality of each study.

## Data analysis

Due to heterogeneity among selected studies, formal quantitative synthesis meta-analysis was not conducted.

# RESULTS

## Study selection

A preliminary database search yielded 750 papers. A manual search resulted in the discovery of five papers. After eliminating duplicate records, 446 records matched the inclusion criterion. Based on the exclusion criteria, 432 papers were eliminated after screening the abstracts. Finally, 14 papers were eligible for qualitative analysis. Reasons for excluding the studies are depicted in the PRISMA as shown in Fig. 2. Detailed characteristics of each study are depicted in Table 1.

## Study quality assessment

We graded all the selected papers after the critical appraisal was completed. Eleven of them were rated as papers of strong evidence ranged from 70% to 98.33%. Three papers were rated as papers of moderate evidence ranged from 61.66% to 65%. None of the studies were with weak evidence as shown in Table 2.

Two authors performed the quality appraisal of the included papers (EAA and NHA). For RCT, the quality of the included studies were assessed using the Cochrane Risk of Bias tool (RoB2) (*Sterne et al., 2019*). A total of five domains are examined for the RoB2 test, with judgments ranging from minimal risk of bias to some concerns and to high risk of bias. The overall risk of bias usually corresponds to the worst risk of bias in any of the domains (Fig. 3). For non-randomized clinical trials, two authors (MO and RS) used ROBINS-I instrument to assesses a total of seven domains, with low risk, moderate risk, severe risk, and critical risk of bias being the judgments. The low risk of ROBINS-I corresponds to a high-quality non-randomized study. Overall, for low risk, the study is judged to be at low risk of bias for all domains; for moderate risk, the study is judged to be at low/moderate risk of bias for all domains; for serious risk, the study is judged to be at serious risk of bias in at least one domain, but not at critical risk of bias in any domain and for critical risk, the study is judged to be at critical risk of bias in at least one domain (Fig. 4).

*In vitro* studies included in this review were assessed with the tool developed by the United States national toxicology program (*Rooney, 2015*). The tool consists of seven criteria (i) Experimental condition bias; (ii) blinding during study; (iii) incomplete data; (iv) exposure characterization (v) Outcome assessment (vi) Reporting bias (vii) Other.

Abdulhameed et al. (2022), *PeerJ*, DOI 10.7717/peerj.12951

**Table 1  A descriptive summary of the findings are tabulated based on the focused question.**

| # | Author year | Title | Study design | In vivo/in vitro studies cell type | Biomaterial — Material used | Biomaterial Characteristics/Dimensions | Limitations | Assay used | Pathway identified | Osteogenesis/angiogenesis | Outcome/Findings | Quality of study (100) |
|---|---|---|---|---|---|---|---|---|---|---|---|---|
| 1 | (Zhang et al., 2011) China | Analysis of the cytotoxicity of differentially sized titanium dioxide nanoparticles in murine MC3T3-E1 preosteoblasts | Cell culture study | MC3T3-E1 murine pre-osteoblasts | Titanium oxide nanoparticles (TiO₂ NPs) | 5 and 32 nm in diameter | * The cellular and molecular cross talk in bone remodeling were not identified. * The effect of TiO₂NPs induced oxidative stress on the osteogenesis-angiogenesis coupling in bone remodeling were not identified. | The tetrazolium salt MTT method and lactate dehydrogenase (LDH) assay. Annexin V apoptosis detected by a flow cytometric assay. TEM analysis. Mitochondrial membrane permeability assay. RNA extraction and real-time quantitative RT-PCR analysis. | ND | Osteogenesis | TiO₂ NPs induced a time- and dose dependent decrease in cell viability. There was a significant increase in lactate dehydrogenase (LDH) release, apoptosis and mitochondrial membrane permeability following short term exposure of the cells to TiO₂ NPs. Compared with the 32 nm TiO₂ NPs, 5 nm TiO2 NPs were more toxic. | 70 Strong Evidence |
| 2 | (El-Shenawy et al., 2012) Saudi Arabia | Oxidative stress and antioxidant responses of liver and kidney tissue after implantation of titanium or titanium oxide coated plate in rat tibiae | Cell culture study + Animal study | Wistar rats Serum | TiO₂ plate Ti plare | 3.0 × 1.0 × 0.25 mm | * The cellular and molecular cross talk in bone remodeling were not identified. * The effect of TiO₂ nanotubes reduced oxidative stress on the osteogenesis-angiogenesis coupling in bone remodeling were not identified. | Plasma mass spectrometer (ICP-MS). Hematology autoanalyzer Cell counter (Sysmex, model KX21N). Flame photometry. Bio-Rad protein assay reagent. Lipid peroxidation assay. Superoxide dismutase (SOD) activity assay. Reduced glutathione assay. ferric reducing/ antioxidant power (FRAP) assay. | ND | NA oxidative stress | Ti-implantation could not decrease thiobarbituric acid reactive product malondialdehyde (MDA) level. TiO₂/Ti-plate did not induce elevation of MDA in liver and kidney tissues, however, some antioxidant have been changed. TiO₂/Ti-plate has less effect on the redox stat of rat than Ti-plate for use as an artificial surgical implant. | 61.66 Moderate Evidence |
| 3 | (Lee et al., 2013) Korea | Bone regeneration around N-acetyl cysteine-loaded nanotube titanium dental implant in rat mandible | Cell culture study + Animal study | MC-3T3 E1 osteoblast-like cells. Sprague Dawley rats | Ti nanotubes. Ti nanotube mini screws. | 1 × 1 cm 1 mm and 6 × 6 cm 1 mm in vitro. 4.5 mm in length and a diameter of 0.85 mm were used for in vivo. | * The cellular and molecular cross talk in bone remodeling were not identified. * The effect of Ti nanotube reduced oxidative stress on the osteogenesis-angiogenesis coupling in bone remodeling were not identified. | wettability. assessment of surface hydrophobicity and hydrophilicity. Cell viability assays. Total nitric oxide (NO) analysis. ELISA. Western blot analysis. Micro-computed tomography (m-CT) analysis. (H&E) staining and immunohistochemical (IHC) staining | RANKL expression | Osteogenesis | MC-3T3-E1 cells seeded on pure Ti and nanotube Ti surfaces increased the expression of RANKL and markedly diminished the expressions of BMP-2, -7 and SPARC. Compared with NLP-Ti, NLN-Ti surfaces attenuated the level of RANKL expression and expression of antioxidants enzymes and bone formation molecules, such as BMP-2 and -7. | 86.66 Strong Evidence |
| 4 | (Pietropaoli et al., 2013) Italy | Glycation and oxidative stress in the failure of dental implants: a case series | case series | Human Study | Dental Implant | ND | * Limitation of the sample size * The molecular pathways that are involved in periimplantitis were not identified. * Implant characteristics were not mentioned. | SDS-Page Electrophoresis. Western Blotting. Colorimetric assay for ThioBarbituric Acid Reactive Substances (TBARS). | ND | NA oxidative stress | The chronic periodontal disease group showed higher oxidative stress than periimplantitis and healthy groups. Periimplantitis group compared to the healthy one had higher oxidative stress levels | 65 Moderate Evidence |
| 5 | (Xie et al., 2014) ) China | Study on potential toxic of titanium oxide nanoparticles on osteoblasts | Cell culture study | Osteoblast cells | Titanium oxide nanoparticles (TiO2-NPs) | less than 25 nm. | * The cellular and molecular cross talk in bone remodeling were not identified. * The effect of TiO₂NPs induced oxidative stress on the osteogenesis-angiogenesis coupling in bone remodeling were not identified. | MTS reagent kit for cytoactive detection. LDH reagent kit for cytotoxicity detection. ROS detection reagent kit. RIPA cell lysis. Transmission electron microscopy (TEM). Zetasizer particle size analyzer Malvern Instrument, Ultrasonic oscillation instrument. Inverted phase contrast microscope. | ND | Osteogenesis | TiO₂ -NPs can cause the decrease of the survival rate of the osteoblast and increase of the content of the LDH released by the cell, and with dosage dependency effect | 71.66 Strong Evidence |

Abdulhameed et al. (2022), *PeerJ*, DOI 10.7717/peerj.12951

**Table 1** (*continued*)

| # | Author year | Title | Study design | *In vivo/ in vitro* studies cell type | Biomaterial | | Limitations | Assay used | Pathway identified | Osteogenesis/ angiogenesis | Outcome/ Findings | Quality of study (100) |
|---|---|---|---|---|---|---|---|---|---|---|---|---|
| | | | | | Material used | Biomaterial Characteristics/ Dimensions | | | | | | |
| 6 | (*Niska et al., 2015*) Poland | Titanium dioxide nanoparticles enhance production of superoxide anion and alter the antioxidant system in human osteoblast cells | Cell culture study | hFOB 1.19 human osteoblast cells | $TiO_2$ NPs | $TiO_2$ NPs size 5–15 nm | * The cellular and molecular cross talk in bone remodeling were not identified. * The effect of $TiO_2$ NPs induced oxidative stress on the osteogenesis-angiogenesis coupling in bone remodeling were not identified. | Water-soluble tetrazolium salt (WST) 1 assay. Mitochondrial activity assay. Transmission electron microscope (TEM) analysis. Lactate dehydrogenase (LDH) assay. Alkaline phosphatase (ALP) activity. Flow cytometry. Total antioxidant capacity. | SIR3 ROS pathway | Osteogenesis | Significant positive correlation between SIR3 and MnSOD at the protein level and a significant negative correlation between decreased SIR3 protein level and increased $O2^{\bullet-}$ level | 83.33 Strong Evidence |
| 7 | (*Mariarosaria et al., 2017*) Italy | Oxidative Stress Evaluation During Perimplantar Bone Resorption in Immediate Post-Extractive Implant | RCT | Human study | Dental Implant | V3 MIS® implant | * The molecular pathways that are involved in perimplantar bone resorption were not identified. * The cellular and molecular cross talk in bone remodeling were not identified. * The effect of biomaterial induced oxidative stress on the osteogenesis-angiogenesis coupling in bone remodeling were not identified. | Thiobarbituric acid reactive substances (TBARs). Nitrite assay. Immunoenzymatic assay for interleukin. Cicloxygenase-2 (COX-2) immunoprecipitation. | ND | Osteoclastogenesis | An increase, during implant integration between the first and third week, of both oxidative stress markers and cyclooxygenase-2 expression. At sixteen week the parameters evaluated returned to basal values. | 63.33 Moderate Evidence |
| 8 | (*Borys et al., 2018*) Poland | Exposure to Ti4Al4V Titanium Alloy Leads to Redox Abnormalities, Oxidative Stress, and Oxidative Damage in Patients Treated for Mandible Fractures | RCT | Human study | Ti4Al4V Titanium alloy | Gray-pigmented periosteum adhered to the titanium miniplates | Only the most commonly used biomarkers of oxidative stress; therefore were evaluated, the assessment of other parameters may lead to different observations and conclusions. | Antioxidant Assays Total antioxidant capacity (TAC). Oxidative Damage Determination Assay | ND | Osteoclastogenesis | Increased activity/concentration of antioxidants both in the mandibular periosteum and plasma/erythrocytes of patients with titanium mandibular fixations. | 77.66 Strong Evidence |
| 9 | (*Hu et al., 2018*) China | Angiogenesis impairment by the NADPH oxidase-triggered oxidative stress at the bone-implant interface: Critical mechanisms and therapeutic targets for implant failure under hyperglycemic conditions in diabetes | Cell culture study + Animal study | HUVEC | Ti4Al4V Titanium alloy | (1) Circular disks (2) Sscrews. Circular disks were used in experiments *in vitro* and screws in study *in vivo*. | * The cellular and molecular cross talk in bone remodeling were not identified. * The effect of biomaterial induced oxidative stress on the osteogenesis-angiogenesis coupling in bone remodeling were not identified. | Immunofluorescent histochemistry. Real-time quantitative PCR (qPCR). Micro-CT analysis. MTT Assay. Matrigel tube-formation assay. Wound-healing assay. ROS assay. Western blot analysis. Immunohistochemical evaluation. ATP assay. Analysis of mitochondrial membrane potential (MMP). | NOX, APO, Nog and Bmp-2 | osteogenesis, angiogensis and adipogenesis | The advanced glycation end products (AGEs)-related and NOX-triggered cellular oxidative stress leads to vascular endothelial cell (VEC) dysfunction and angiogenesis impairment at the bone-implant interface (BII), which plays a critical role in the compromised implant osteointegration under diabetic conditions. | 90 Strong Evidence |
| 10 | (*Yu et al., 2018*) China | Osteogenesis potential of different titania nanotubes in oxidative stress microenvironment | Cell culture study | Calvaria osteoblasts | Titanium foils | TNT30, TNT70 and TNT110 | * The cellular and molecular cross talk in bone remodeling were not identified. * The effect of $TiO_2$ nantube reduced oxidative stress on the osteogenesis-angiogenesis coupling in bone remodeling were not identified. | Scanning electron microscopy. Atomic force microscopy. X-ray diffraction. Video-based optical system. MTT. BCA assay kit. confocal laser scanning microscopy. quantitative-polymerase chain reaction (q-PCR). | Wnt signals | Osteogenesis | Large nanotubes displayed strong capacities to improve cell adhesion, survival and differentiation of osteoblasts after $H_2O_2$ treatment. | 86.66 Strong Evidence |

Abdulhameed et al. (2022), *PeerJ*, DOI 10.7717/peerj.12951

**Table 1** (*continued*)

| # | Author year | Title | Study design | *In vivo/ in vitro* studies cell type | Biomaterial Material used | Biomaterial Characteristics/ Dimensions | Limitations | Assay used | Pathway identified | Osteogensis/ angiogenesis | Outcome/ Findings | Quality of study (100) |
|---|---|---|---|---|---|---|---|---|---|---|---|---|
| 11 | (*Shen et al., 2019*) China | Titania nanotubes promote osteogenesis via mediating crosstalk between macrophages and MSCs under oxidative stress | Cell culture study | Mesenchymal stem cells (MSCs). RAW264.7 cells. | Titanium foils. Titanium nanotubes | TNT30, TNT70 and TNT110 | Should be implemented *in vivo* and in humans. | Confocal laser scanning microscope. Cell viability assay. Nitrous oxide (NO) assay ELISA. real-time quantitative-polymerase chain reaction (q-PCR) technique. Western Blot (WB) technique | integrin/FAK-mediated MAPK and NF$\kappa$B signals | Osteogenesis | Large nanotubes (110 nm) could recruit more MSCs to the injury site than Ti and TNT30 substrates by increasing the chemokine expressions of RAW264.7 cells under OS. | 99.33 Strong Evidence |
| 12 | (*Mijiritsky et al., 2019*) Italy | Presence of ROS in inflammatory environment of peri-implantitis tissue: *in vitro* and *in vivo* human evidence | Cross-sectional study | Human study | Dental Implant | ND | * Limitation of the sample size * The molecular pathways that are involved in periimplantitis were not identified. * Implant characterestics were not mentioned. | Immunohistochemistry and Histomorpholgical Analyses. Immunofluorescence Staining. Transmission Electron Microscopy (TEM). RNeasy Mini Kit. | WNT, HEDGE-HOG, and FOXO | Osteogenesis and adipogenesis | Osteogenesis down expressed in peri-implantitis and up regulated in the control. | 78.33 Strong Evidence |
| 13 | (*Yang et al., 2020*) China | TiO$_2$ Nanotubes Alleviate Diabetes-Induced Osteogenetic Inhibition | Cell culture study + Animal study | pre-osteoblastic cell line, MC3T3-E1 Sprague–Dawley (SD) rats. | *In vitro*; Ti discs divided into 3 groups –mechanically polished (MP group), sandblasted and acidetched (SLA group), and oxidized TiO$_2$ nanotubes (TNT group). *In vivo*; pure Ti implants | *In vitro*; dimensions: $10.0 \times 10.0 \times 0.3$ mm$^4$ or $20.0 \times 20.0 \times 0.3$ mm$^4$. *In vivo*; a diameter and length of 2 and 4 mm. | * The cellular and molecular cross talk in bone remodeling were not identified. * The effect of TiO$_2$ nantube reduced oxidative stress on the osteogenesis-angiogenesis coupling in bone remodeling were not identified. | CCK-8 assay. Alkaline phosphatase (ALP) assay. osteopontin (OPN) assay. Western blot test. Alizarin Red staining. Flow cytometry. Superoxide Dismutase (SOD) activity sssay. micro-CT scan. | ND | Osteogenesis | High-glucose conditions inhibited ALP and OPN expressions on different modified Ti surfaces, TNT surface could alleviate the inhibition of ALP and OPN expressions under high-glucose conditions. High-glucose conditions inhibited osteogenesis on different modified Ti surfaces, The TNT surface could alleviate the inhibition of mineralization when compared with the SLA surface under high glucose conditions. | 88.33 Strong Evidence |
| 14 | (*Huang et al., 2021*) China | Bioadaptation of implants to *in vitro* and *in vivo* oxidative stress pathological conditions via nanotopography-induced FoxO1 signaling pathways to enhance Osteoimmunal regeneration | Cell culture study + Animal study | 1) BMSCs 2) RAW264.7 | TiO$_2$ nanotube (TNT). | 1) Pure Ti foils 2) Cylinder- shaped (disk)pure Ti implants. | * The cellular and molecular cross talk in bone remodeling were not identified. * The effect of biomaterial induced oxidative stress on the osteogenesis-angiogenesis coupling in bone remodeling were not identified. | Scanning electron microscopy. Laser scanning confocal microscope profilometer. Contact angle analyzer. | FoxO1-induced oxidation resistance and anti-inflammatory osteoimmunity. | Osteogenesis | Nanoscale TNT coatings on titanium implants exhibited superior osteogenesis and osseointegration compared with microscale SLA surfaces. | 86.66 Strong Evidence |

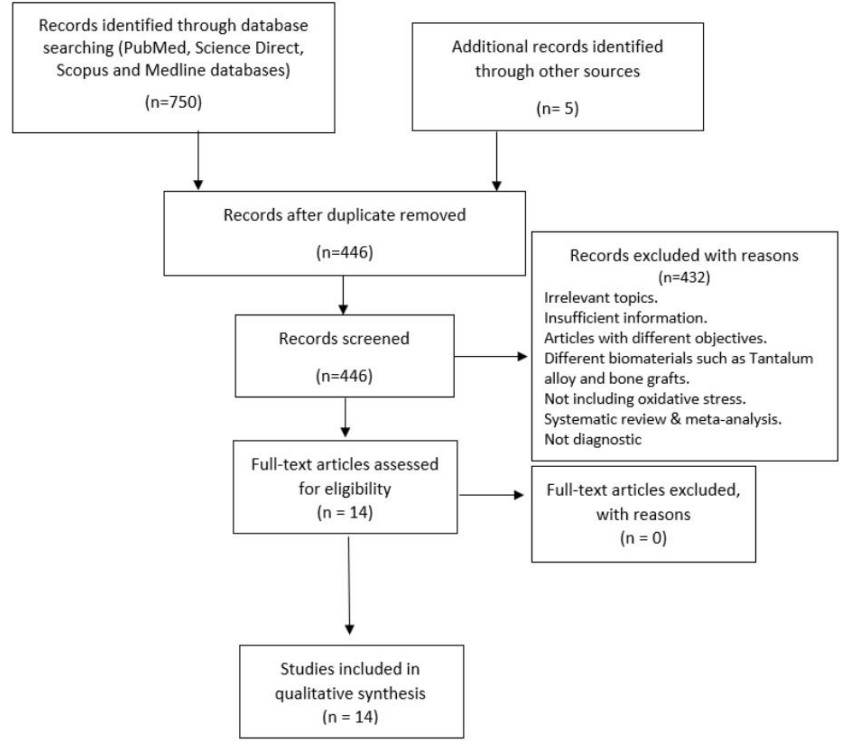

**Figure 2** **Summary of the systematic review workflow using PRISMA chart.**

The interpretation for fulfilling a "moderate", "low", and "no information" score was described in Fig. 5.

## Studies characteristics

All the studies that were included took place between the years of 2000 and 2021. Of the fourteen studies, five publications were undertaken *in vitro*; most of them were performed in China (*Zhang et al., 2011*; *Xie et al., 2014*; *Yu et al., 2018*; *Shen et al., 2019*) and only one was performed in Poland (*Niska et al., 2015*). Other five studies were conducted *in vitro/vivo*, three of them performed in China (*Hu et al., 2018*; *Yang et al., 2020*; *Huang et al., 2021*), One in Saudi Arabia (*El-Shenawy et al., 2012*) and one in Korea (*Lee et al., 2013*). The last four studies were done in humans and, three of them performed in Italy (*Pietropaoli et al., 2013*; *Mariarosaria et al., 2017*; *Mijiritsky et al., 2019*) and one in Poland (*Borys et al., 2018*). All these fourteen studies studied the effect of titanium dioxide nanoparticles, nanotubes and titanium alloy on the generation of ROS and osteogenesis along with the relevant affected signaling pathways such as Wnt, MAPK and NOX pathway.

## Study outcome

The fourteen articles that were chosen did not all shed the light on the same type of titanium dioxide, in fact 6 of those (*El-Shenawy et al., 2012*; *Lee et al., 2013*; *Yu et al., 2018*; *Shen et al., 2019*; *Yang et al., 2020*; *Huang et al., 2021*) focused on of titanium dioxide nanotube effect on oxidative stress and osteogenesis and 3 studies (*Zhang et al., 2011*; *Xie et al., 2014*;

Abdulhameed et al. (2022), *PeerJ*, DOI 10.7717/peerj.12951

**Table 2   Critical appraisal skills program checklist for quality assessment of observational studies (CASP) (*Long, French & Brooks, 2020*).**

| NO. | Study authors | Critical appraisal of introduction 16.65% | | | Critical appraisal of methodology 26.64% | | | | | | | | | | Critical appraisal of results and discussion 56.61% | | | | | | | | | | | | | | | | | CASP score 100% |
|---|---|---|---|---|---|---|---|---|---|---|---|---|---|---|---|---|---|---|---|---|---|---|---|---|---|---|---|---|---|---|---|---|---|
| 1 | (*Zhang et al., 2011*) (China) | √ | √ | √ | √ | X | √ | X | √ | √ | X | √ | √ | X | √ | X | √ | √ | X | √ | X | √ | X | X | √ | X | √ | √ | √ | √ | √ | 70 Strong Evidence |
| 2 | (*El-Shenawy et al., 2012*) (Saudi Arabia) | √ | √ | √ | √ | X | √ | X | √ | √ | X | √ | √ | X | √ | X | √ | √ | X | √ | X | √ | X | √ | X | √ | √ | | | | | 61.66 Moderate Evidence |
| 3 | (*Lee et al., 2013*) (Korea) | √ | √ | √ | √ | √ | √ | √ | √ | √ | √ | √ | √ | √ | √ | √ | √ | √ | X | √ | √ | √ | √ | X | X | √ | √ | √ | X | | | | 86.66 Strong Evidence |
| 4 | (*Pietropaoli et al., 2013*) (Italy) | √ | √ | √ | √ | X | √ | X | √ | √ | X | √ | √ | X | √ | √ | √ | X | √ | X | X | √ | √ | √ | √ | X | √ | √ | | | | | 65 Moderate Evidence |
| 5 | (*Xie et al., 2014*) (China) | √ | √ | √ | √ | X | √ | X | √ | √ | X | √ | √ | √ | √ | √ | X | √ | X | √ | X | X | √ | √ | √ | √ | √ | √ | | | | | 71.66 Strong Evidence |
| 6 | (*Niska et al., 2015*) (Poland) | √ | √ | √ | √ | √ | √ | X | √ | √ | √ | √ | √ | √ | √ | √ | X | √ | X | √ | √ | X | √ | X | √ | √ | √ | √ | | | | | 83.33 Strong Evidence |
| 7 | (*Mariarosaria et al., 2017*) (Italy) | √ | √ | √ | √ | X | √ | X | √ | √ | X | √ | √ | X | √ | √ | √ | X | √ | X | √ | X | √ | X | √ | X | √ | √ | | | | | 63.33 Moderate Evidence |
| 8 | (*Borys et al., 2018*) (Poland) | √ | √ | √ | √ | √ | √ | X | √ | √ | √ | √ | X | √ | √ | √ | X | √ | √ | √ | √ | X | √ | X | X | √ | √ | √ | | | | | 77.66 Strong Evidence |
| 9 | (*Hu et al., 2018*) (China) | √ | √ | √ | √ | √ | √ | √ | √ | √ | √ | √ | √ | √ | √ | √ | √ | X | √ | √ | √ | √ | √ | X | √ | √ | √ | | | | | | 90 Strong Evidence |
| 10 | (*Yu et al., 2018*) (China) | √ | √ | √ | √ | √ | √ | √ | √ | √ | √ | √ | √ | √ | √ | √ | X | √ | √ | √ | √ | X | X | √ | √ | √ | √ | | | | | | 86.66 Strong Evidence |
| 11 | (*Shen et al., 2019*) (China) | √ | √ | √ | √ | √ | √ | √ | √ | √ | √ | √ | √ | √ | √ | √ | √ | √ | √ | √ | √ | √ | √ | √ | √ | √ | √ | | | | | | 98.33 Strong Evidence |
| 12 | Mijiritsky et al., 2019 (Italy) | √ | √ | √ | √ | √ | √ | X | √ | √ | √ | √ | X | √ | √ | √ | X | √ | X | √ | √ | √ | √ | √ | √ | X | √ | √ | √ | | | | 78.33 Strong Evidence |
| 13 | (*Yang et al., 2020*) (China) | √ | √ | √ | √ | √ | √ | √ | √ | √ | √ | √ | √ | √ | √ | √ | √ | √ | √ | X | √ | √ | √ | √ | X | √ | √ | √ | √ | | | | 88.33 Strong Evidence |
| 14 | (*Huang et al., 2021*) (China) | √ | √ | √ | √ | √ | √ | √ | √ | √ | √ | √ | √ | √ | √ | √ | √ | √ | √ | √ | √ | X | √ | √ | √ | X | X | √ | √ | √ | √ | | 86.66 Strong Evidence |

**Notes.**

√, Point awarded; X, Point not awarded.

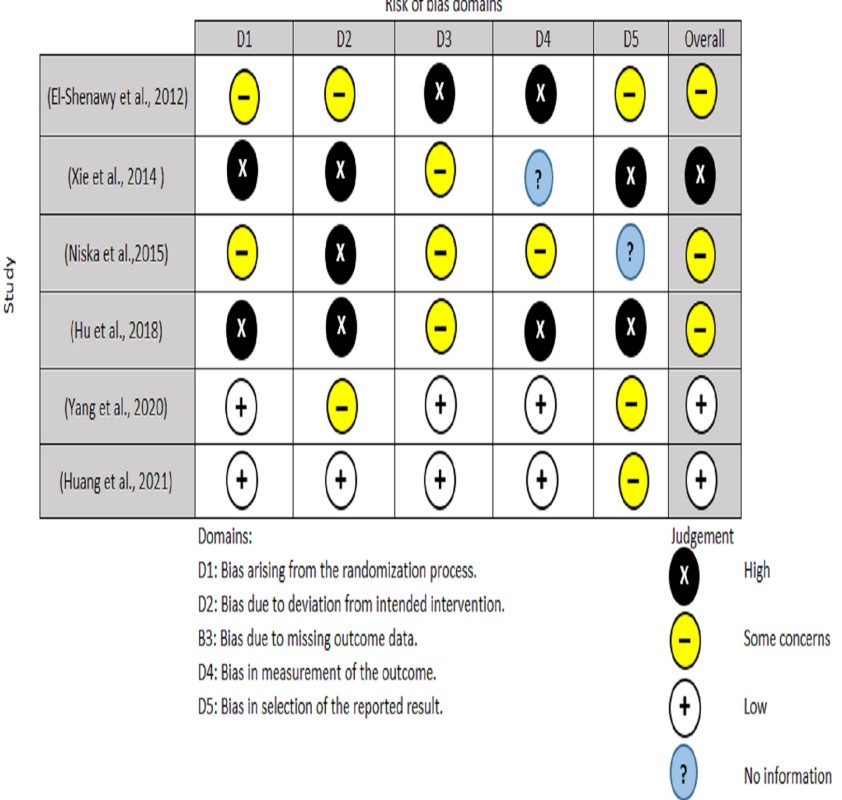

**Figure 3** Risk bias assessment for randomized clinical trail.

*Niska et al., 2015*) focused on the effect of titanium dioxide nanoparticles on oxidative stress and osteogenesis therefore, highlighting theses results were done first.

### TNT effect on oxidative stress
It was found that TiO$_2$ nanotube (TNT) coating on titanium implants is directly inducing superior osteogenic differentiation of bone mesenchymal stem cells (MSCs) and osseointegration compared with microscale sand blasted-acid etched topography (SLA) (*Huang et al., 2021*). *Huang et al. (2021)* in their study found that the increased forkhead box transcription factor O1 (FoxO1) drives oxidation resistance on TNT during oxidative stress (OS) and TNT decreases oxidative stress (OS) in macrophages indirectly, leading in a higher proportion of the M2 phenotype under OS and increased secretion of the antiinflammatory cytokine IL-10 could improves osseo-immunity capability in contrast to SLA. Shen et al. in their study on cell culture confirmed that large nanotubes (110 nm) were shown to greatly aggravate early inflammatory responses of RAW264.7 cells by up-regulating ITG-mediated MAPK and NF$\kappa$B pathways. This further increased the expressions of osteogenesis and chemokine genes like BMP2, VEGF, IL-8, SDF1 and CCL2 (*Shen et al., 2019*). Co-culture of RAW264.7 with MSCs results in more MSCs that were effectively recruited by inflammatory RAW264.7 cells on TNT110 substrates which secretes many antiinflammatory cytokines like such as IL-4, IL-10, IL-13, and TGF $\beta$1 which

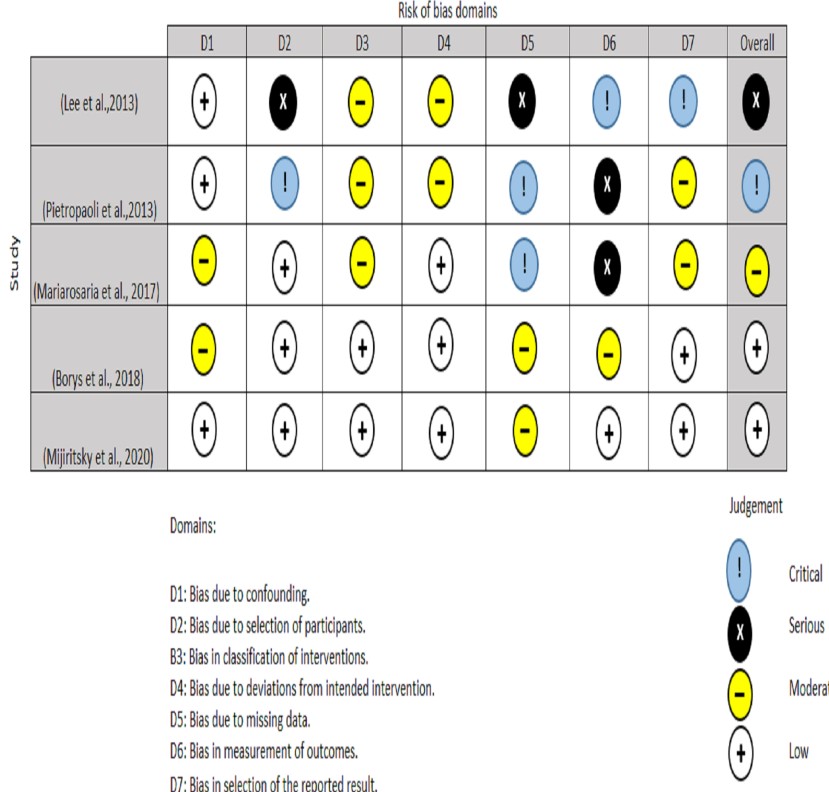

**Figure 4** Risk bias assessment for non-randomized clinical trials.

suppresses inflammation of RAW264.7 cells by increasing M1-to-M2 transition. Crosstalk between MSCs and RAW264.7 cells caused by 110 nm nanotubes was found to be the key factor promoting early osteogenic induction in the TNT110 group (*Shen et al., 2019*). Yu et al. in their study on calvaria osteoblasts seeded onto different substrates investigated the anti-oxidative properties of various $TiO_2$ nanotubes (TNTs) to screen the desirable size for improved steogenesis and reveal the underlying molecular mechanism. He discovered that oxidative stress resistance of large $TIO_2$ nanotubes was linked to high expression of integrin 51 (ITG 51), which up-regulated the production of anti-apoptotic proteins (p-FAK, p-Akt, p-FoxO3a, and Bcl2) while down-regulating the expression of pro-apoptotic proteins (p-FAK, p-Akt, p-FoxO3a (Bax). Wnt signals on the other hand like Wnt3a, Wnt5a, Lrp5, Lrp6, and -catenin have also been discovered to play a function in encouraging osteogenic differentiation of osteoblasts in oxidative stress condition (*Yu et al., 2018*). El-Shenawy et al. used $TiO_2$ nanotube and Ti-plate in rats as an artificial surgical implant, they found that $TiO_2$ nanotube did not induce elevation of MDA in liver and kidney tissues, however, some antioxidant have been changed. However, $TiO_2$ nanotube has less effect on the redox state of rat than Ti-plate for use as an artificial surgical implant (*El-Shenawy et al., 2012*). In 2013, Lee et al., examined the effect of N-acetyl cysteine (NAC)-loaded nanotube titanium (NLN-Ti) implants on antioxidants enzymes and bone formation. MC-3T3-E1

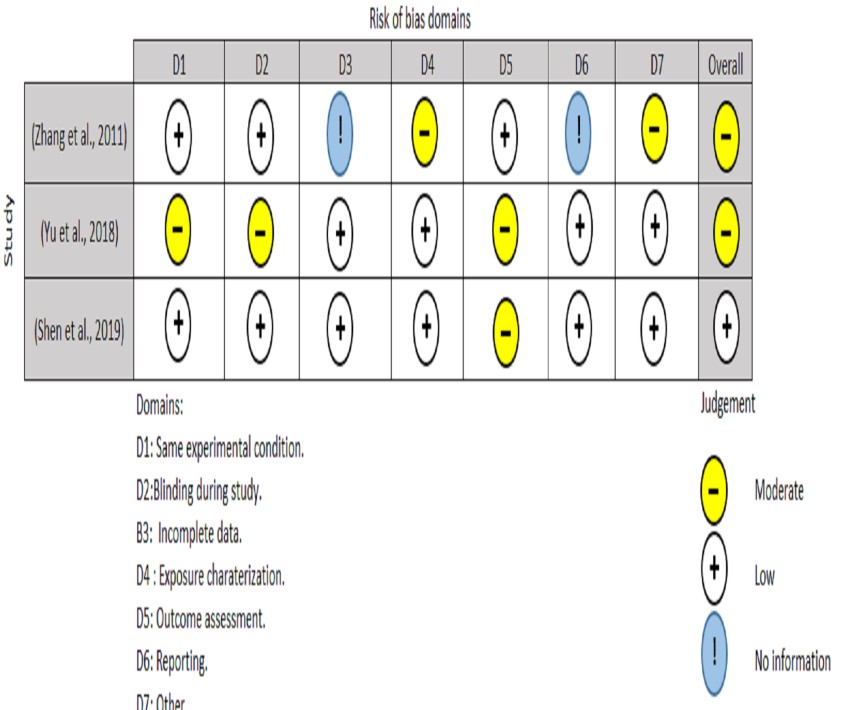

**Figure 5   Risk of bias assessment of *in vitro* studies.**

cells were seeded on pure Ti and nanotube Ti surfaces increased the expression of RANKL and markedly diminished the expressions of bone morphogenic protein BMP-2, -7. Lee et al. found that NAC-loaded nanotube Ti (NLN-Ti) surfaces attenuated the level of RANKL expression and expression of antioxidants enzymes and bone formation molecules, such as BMP-2 and -7.(19) Yang et al. also investigated if titanium implants with $TiO_2$ nanotubes (TNT) surface can retain their biocompatibility and osteogenetic ability under diabetic conditions in rats, they found that high-glucose conditions in diabetic rats inhibited alkaline phosphatase (ALP) and osteopontin (OPN) on different modified Ti surfaces, as TNT surface could alleviate the inhibition ALP and OPN expressions under high-glucose conditions. High-glucose conditions inhibited osteogenesis on different modified Ti surfaces, The TNT surface could alleviate the inhibition of mineralization when compared with the SLA surface under high glucose conditions (*Yang et al., 2020*).

### TiO₂NP effect on oxidative stress

In 2015, *Niska et al. (2015)* subjected hFOB 1.19 cells to $TiO_2$NPs and found that $TiO_2$NPs significantly reduce the levels of SIR3 protein (sirtuin which is an enzyme involved in many cellular processes, it protects cells against stress and control a number of metabolic pathways) in osteoblast cells. They found a significant positive correlation between SIR3 and MnSOD at the protein level and a significant negative correlation was found between decreased SIR3 protein level and increased superoxide ($O_2^{\bullet-}$) level. They concluded that $TiO_2$NP-could induce toxicity in osteoblast cells (*Niska et al., 2015*).

MC3T3-E1 murine preosteoblasts seeded on TiO$_2$ NPs of 5 and 32 nm in diameter to assess the cytotoxic effects of TiO$_2$ nanoparticles (NPs) of different sizes on murine MC3T3-E1 preosteoblasts. Therefore several tests were conducted like cell viability and cytotoxicity assays, flow cytometry, TEM analysis and quantitative RT-PCR assay. After exposure of MC3T3-E1 murine pre-osteoblast cells to TiO$_2$-NPs, the the survival rate of the osteoblast decreased and the content of the lactate dehydrogenase (LDH) released by the cell increased (*Zhang et al., 2011*; *Xie et al., 2014*).

### Other titanium alloy effect on oxidative stress

*Borys et al. (2018)* performed a study to evaluate the influence of Ti6Al4V titanium alloy on redox balance and oxidative damage in the periosteum surrounding the titanium miniplates and screws as well as in plasma and erythrocytes of patients with mandibular fractures. In their study they found that the occurrence of redox imbalance as well as oxidative stress and oxidative damage in the periosteum surrounding the Ti6Al4V titanium alloy has increased, as well as activity/concentration of antioxidants both in the mandibular periosteum and plasma/erythrocytes of patients with titanium mandibular implants (*Borys et al., 2018*).

### Effect of local and systemic factors on oxidative stress around implants

*Pietropaoli et al. (2013)* investigated the presence of the Advanced Glycation End products (AGEs) and oxidative stress in periimplantitis, they compared subjects with chronic periodontitis and periimplantitis to healthy subjects and they found that subjects with periodontitis had significantly higher oxidative stress than periimplantitis and healthy groups, and that subjects with periimplantitis had significantly higher oxidative stress than healthy subjects. To confirm the presence of ROS in inflammatory meliue of peri-implantitis tissue, Mijiritsky et al., in 2019, conducted a cross-sectional study on patients with peri-implantitis (in which circumferential peri-implant soft tissue samples were collected during respective surgical treatment of peri-implantitis or in case of extraction of failed implants due to peri-implant disease) and patients with healthy periodontium (a specimen of mucosa was collected from the healing abutment at the second stage of implant uncovery). They suggested that peri-implantitis lesions exhibit a well defined biological organization not only in terms of inflammatory cells but also on vessel and extracellular matrix components even if no difference in the epithelium is evident, and the presence of reactive oxygen species (ROS) related to the inflammatory environment influences the correct commitment of mesenchymal stem cells as confirmed by Immunohistochemistry and histomorphglogical analyses, immunofluorescence staining and Transmission Electron Microscopy (TEM) (*Mijiritsky et al., 2019*).

*Mariarosaria et al. (2017)* evaluated the presence of oxidative stress during peri-implant bone resorption in immediate post-extractive implant, they found an increase in both oxidative stress markers and cyclooxygenase-2 expression during implant integration for the period between the first and third week, however at sixteenth week the parameters evaluated returned to basal values. Finally, *Hu et al. (2018)* highlighted the role of angiogenesis in the diabetes-induced poor bone repair at the bone-implant interface (BII) and the related mechanisms. They suggested that the advanced glycation end products (AGEs)-related and NOX-triggered cellular oxidative stress leads to vascular endothelial cell (VEC) dysfunction

and angiogenesis impairment at the bone-implant interface (BII), which plays a critical role in the compromised implant osteointegration under diabetic conditions.

## DISCUSSION

High quality and quantity of bone regeneration is the ultimate aim of an implanted biomaterial. Surgical trauma leading to inflammation is natural sequalae during implantation procedure generating ROS that is necessary to drive multiple signal transduction for molecular healing process (*Yanez, Blanchette & Jabbarzadeh, 2017*). Very often the site of implantation needing repair has already been infected and a certain degree of ROS production has already pre-existed even before surgical implantation of biomaterial (*Mouthuy et al., 2016*). Combination of pre-existing ROS together with postoperative implantation trauma may produce an amount of oxidative stress that may exceed the antioxidant capacity in that particular site. This challenges may lead to inadequate bone implant integration and without further support with anti-inflammatory and antibiotics and adjustment of loading forces may lead to failure of osseointegration (*Yanez, Blanchette & Jabbarzadeh, 2017*). However, the mechanism regulating the interaction between ROS and peri-implant environment with respect to producing ideal integration is poorly understood. Various molecular processes that interact with the biomaterial surface topography, peri-implant tissues, angiogenesis and antioxidants measures have been implicated in this systematic review.

The communication between bone-forming osteoblasts and bone resorbing osteoclasts as well as vascular endothelial cells is a fundamental requirement for effective and balanced bone remodeling (*Yin et al., 2021*). For biomaterial research in manufacturing novel implant, development of *in vitro* models is necessary to investigate this communication.

The foreign body response to biomaterials is a cascade of events triggered by implantation, followed by protein adsorption, adhesion and activation of immune cells, and ultimately recruitment of fibroblasts and formation of a fibrous capsule (*Liu et al., 2011*). During this process, it is thought that reactive oxygen species (ROS) are released by activated phagocytes, and contribute to oxidative degradation of materials (*Liu et al., 2011*). In this systematic review, most studies found that dental implant enhances ROS generation and reduces osteoblastic activity (*Zhang et al., 2011*; *Pietropaoli et al., 2013*; *Xie et al., 2014*; *Niska et al., 2015*; *Mariarosaria et al., 2017*; *Borys et al., 2018*; *Hu et al., 2018*; *Mijiritsky et al., 2019*). Interestingly, some studies have stated that implant nanotubes could act as antioxidant and promote osseogenesis (*El-Shenawy et al., 2012*; *Lee et al., 2013*; *Yu et al., 2018*; *Shen et al., 2019*; *Yang et al., 2020*; *Huang et al., 2021*). However, this area requires further research.

*Garret et al. (1990)* first described the relationships between oxygen-derived free radicals (particularly the superoxide anion) and the formation and activation of osteoclasts. These findings were confirmed by the study of Lee et al. in 2005 who demonstrated that the receptor activator of nuclear factor-kappaB ligand (RANKL)-induced osteoclastogenesis requires ROS production. Bone marrow (BM) precursor cells was used as an osteoclast differentiation model (*Bai et al., 2005*). Similarly, osteoblasts play important roles in

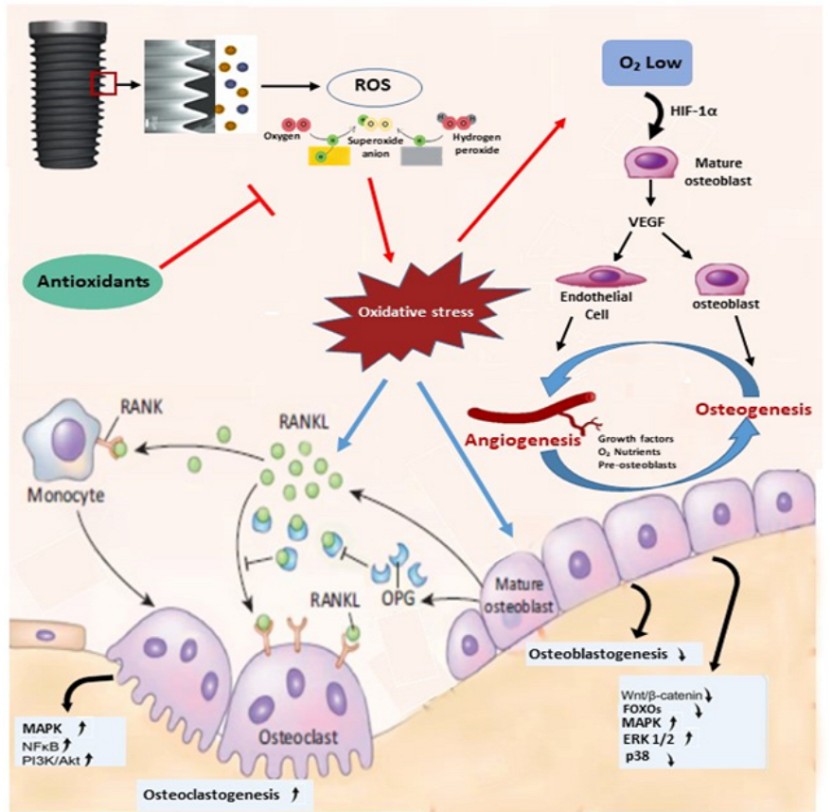

**Figure 6** Schematic figure of the link between dental implant surface, presence of particles release, oxidative stress generation, antioxidant production, and their effects on bone remodeling markers.

osteoclastogenesis through regulating receptor activated nuclear factor kappa B (RANK) ligand (RANKL) and osteoprotegerin (OPG) expression. It was hypothesized that OPG plays an important role in the crosstalk between osteoclasts and osteoblasts in response to biomaterial implantation. The evidences from the *in vitro* and *in vivo* studies suggested that OPG played an important role in the uncoupling effect of biomaterial on host bone cells metabolism, possibly by acting as a cross-talk molecule between osteoclasts and osteoblasts in response to biomaterial implantation (Fig. 6).

Recent studies reporting on co-cultures of osteoblasts and osteoclast used different cell combinations (*Borciani et al., 2020*). Some authors report on successful cultures of osteoblast cell lines or primary osteoblasts in combination with peripheral blood mononuclear cell (PBMC) or isolated monocytes. The cells were of murine or human origin and were cultivated with or without the addition of M-CSF and RANKL (*Borciani et al., 2020*). However, there is not much work in regenerative medicine dealing with co-cultures for investigating the impact of ROS secondary to biomaterial are known.

Oxidative stress can affect osteogenesis-angiogenesis coupling by its effect on VEGF signalling. *Y et al. (2013)* VEGF exerts its action through binding to VEGF Receptor-2 (VEGFR-2, also known as FLK1/KDR) in endothelial cells (ECs), causing

autophosphorylaion of EC in its cytoplasmic tyrosine residues and driving downstream pathway such as PI3K/AKT and MAPK which promote EC proliferation and migration. VEGF stimulates ROS production via Rac-1-mediated NADPH oxidase activation (*Y et al., 2013*) and also increases mitochondria-derived $H_2O_2$ (*Y et al., 2013*).

Based on the above results, there were conflicting findings of the effects of titanium dioxide dental implants on osteogenesis-angiogenesis coupling. Some reports says that TNT enhances OS and impairs osteogenesis -angiogenesis coupling and others says that TNT reduces OS. Studies with strong evidence (*Yu et al., 2018*; *Shen et al., 2019*; *Huang et al., 2021*) were in favor to use titanium dioxide nanotube to reduce oxidative stress and promote osteogenesis-angiogenesis coupling, through activation of Wnt signalling pathways as depicted in Fig. 6.

Limitations of this review were the small sample sizes and the few clinical trials in most studies, which raises the concern over the reliability of the results. Secondly, the molecular pathways that were affected by dental implant induced-oxidative stress and the effect of oxidative stress on the osteogenesis-angiogenesis coupling in bone remodeling were not identified in some studies. Additionally, there was a lack of proper statistical tests in some studies, making determining the significance of ROS generation difficult.

## CONCLUSION

Titanium dioxide nanotube (TNT) can reduce oxidative stress and promote osteoblastic activity through its effect on Wnt, MAPK and FoxO1 signaling pathways. Current scientific evidence is inclined towards supporting the use of $TiO_2$ nanotube-coated titanium implants to reduce oxidative stress and promote osteogenesis. However, more well-designed large sample sized randomized controlled clinical trials are necessary to support our conclusion.

### Funding
The authors received no funding for this work.

### Competing Interests
The authors declare there are no competing interests.

### Author Contributions
- Elaf Akram Abdulhameed, Natheer H. Al-Rawi, Marzuki Omar, Nadia Khalifa and A.B. Rani Samsudin conceived and designed the experiments, performed the experiments, analyzed the data, prepared figures and/or tables, authored or reviewed drafts of the paper, and approved the final draft.

### Data Availability
The PRISMA checklist and the Rationale are available in the Supplementary Files.

## Supplemental Information

Supplemental information for this article can be found online at http://dx.doi.org/10.7717/peerj.12951#supplemental-information.

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
