# Peer review of "Titanium dioxide dental implants surfaces related oxidative stress in bone remodeling: a systematic review"

_PeerJ, doi:10.7717/peerj.12951_

## Round 0.1 · original submission · Major Revisions

Dear authors,
Please follow the reviewers' instructions.

Best regards

Reviewer 1 ·

Basic reporting

The article is well written and complete from different aspects. Data are well reported. I think can be improved in the introduction to the PICO and about the importance of this Review.

Experimental design

The article is a systematic review, the reporting items was correcty followed.

Validity of the findings

the findings from this research are something new in literature, I think that is relevant for its contents.

Annotated reviews are not available for download in order to protect the identity of reviewers who chose to remain anonymous.

Reviewer 2 ·

Basic reporting

Dear Authors,
Thank you for submitting your article to this prestigious journal.
The manuscript is well presented but it presents some issues:

Abstract: Abstract should be revised and make it more appealing.

Keywords: To ensure a properly research in medical databases, use MeSH terms to find appealing keywords that could be helpful in finding your article.

“From Line 46 to 50“ More citations are needed.

From 61 to 65 i would like to suggest the author to provide a figure describing the whole bone remodeling process (describing all the proteins take into account).

From 67 to 68 Citations are needed? What articles??????

Introduction should be revised to focus the attention to the AIM of this manuscript. Moreover introduction is too short please extend it.
AIM should be more appealing
Materials and Methods section is well written and describes a good job.
From Line 324 to 330 It is not clear what authors would comunicate. More clarification regarding ROS activities are needed.
Discussion seems a repetition of the RESULT section please modify
Conclusion must be entirely revised making it more appealing and expanded.
English spell revision is necessary.

Best Regards.

Experimental design

Dear Authors,
Thank you for submitting your article to this prestigious journal.
The manuscript is well presented but it presents some issues:

Abstract: Abstract should be revised and make it more appealing.

Keywords: To ensure a properly research in medical databases, use MeSH terms to find appealing keywords that could be helpful in finding your article.

“From Line 46 to 50“ More citations are needed.

From 61 to 65 i would like to suggest the author to provide a figure describing the whole bone remodeling process (describing all the proteins take into account).

From 67 to 68 Citations are needed? What articles??????

Introduction should be revised to focus the attention to the AIM of this manuscript. Moreover introduction is too short please extend it.
AIM should be more appealing
Materials and Methods section is well written and describes a good job.
From Line 324 to 330 It is not clear what authors would comunicate. More clarification regarding ROS activities are needed.
Discussion seems a repetition of the RESULT section please modify
Conclusion must be entirely revised making it more appealing and expanded.
English spell revision is necessary.

Best Regards.

Validity of the findings

Dear Authors,
Thank you for submitting your article to this prestigious journal.
The manuscript is well presented but it presents some issues:

Abstract: Abstract should be revised and make it more appealing.

Keywords: To ensure a properly research in medical databases, use MeSH terms to find appealing keywords that could be helpful in finding your article.

“From Line 46 to 50“ More citations are needed.

From 61 to 65 i would like to suggest the author to provide a figure describing the whole bone remodeling process (describing all the proteins take into account).

From 67 to 68 Citations are needed? What articles??????

Introduction should be revised to focus the attention to the AIM of this manuscript. Moreover introduction is too short please extend it.
AIM should be more appealing
Materials and Methods section is well written and describes a good job.
From Line 324 to 330 It is not clear what authors would comunicate. More clarification regarding ROS activities are needed.
Discussion seems a repetition of the RESULT section please modify
Conclusion must be entirely revised making it more appealing and expanded.
English spell revision is necessary.

Best Regards.

Additional comments

Dear Authors,
Thank you for submitting your article to this prestigious journal.
The manuscript is well presented but it presents some issues:

Abstract: Abstract should be revised and make it more appealing.

Keywords: To ensure a properly research in medical databases, use MeSH terms to find appealing keywords that could be helpful in finding your article.

“From Line 46 to 50“ More citations are needed.

From 61 to 65 i would like to suggest the author to provide a figure describing the whole bone remodeling process (describing all the proteins take into account).

From 67 to 68 Citations are needed? What articles??????

Introduction should be revised to focus the attention to the AIM of this manuscript. Moreover introduction is too short please extend it.
AIM should be more appealing
Materials and Methods section is well written and describes a good job.
From Line 324 to 330 It is not clear what authors would comunicate. More clarification regarding ROS activities are needed.
Discussion seems a repetition of the RESULT section please modify
Conclusion must be entirely revised making it more appealing and expanded.
English spell revision is necessary.

Best Regards.

Reviewer 3 ·

Basic reporting

ok

Experimental design

ok
metanalysis lacks. I strongly suggest that the authors add it

Validity of the findings

moderate

Additional comments

I would like to congratulate the Authors for the efforts in writing the manuscript. The systematic review is well conducted. However, some improvements are required.
Why did not the Authors perform a metanalysis? Kindly explain the reason.
ABSTRACT
• LINES 30-32: To validate the cellular and molecular cross talk in bone remodeling of the present review. A well-controlled clinical trials with a large sample size are required.
Kindly check the punctuation and remove the dot between sentences.
• Kindly rewrite the keywords in alphabetical order.

RESULTS
• Lines 155: kindly check this typo “nThree papers were rated as”
• Line 180-181: “Of the fourteen studies, five of the fourteen publications were undertaken in vitro;” Kindly correct these sentences “of the fourteen studies” was repeated twice.

DISCUSSION
• Lines 295-307: all those statements should be supported by evidence. Kindly add references in order to support what explained.
• Line 362: “promote osteogenesis-angiogenesis coupling. Through activation of Wnt signalling pathways as depicted in figure”. Kindly check punctuation.
• Kindly add a comment on the future perspectives and suggestions.

---

## Round 0.2 · accepted · Accept

Thank you for submitting your manuscript. The article is now ready for publication.

Reviewer 1 ·

Basic reporting

The article is now suitable for publication.

Experimental design

The systematic review is well conducted, the applied methods are correct.

Validity of the findings

The findings are something new in literature about this topic.

Additional comments

Thank you for the hard work in improving this manuscript.
In my opinion, It's now suitable for publication.

Reviewer 3 ·

Basic reporting

ok

Experimental design

ok

Validity of the findings

ok

Additional comments

Thank you for submitting your manuscript. The article is now ready for publication.

Best regards